# Past, Present, and Future Arctic Radiative States Simulated by Polar-WRF

Cameron Bertossa<sup>1</sup>, Tristan L'Ecuyer<sup>1,2</sup>, and David Henderson<sup>1</sup>

Correspondence: Cameron Bertossa (bertossa@wisc.edu)

Abstract. Two recurring radiative states ("transmissive" and "opaque") strongly modulate the Arctic surface energy balance through their control on downwelling longwave radiation (DLR). Because these states are primarily governed by cloud processes, many coarse-resolution models fail to capture their behavior. This study evaluates how well the Polar-optimized Weather Research and Forecasting model (PWRF) simulates present-day DLR distributions associated with these states and examines projected changes into the future. While most physics parameterizations mirror those of the widely used Arctic System Reanalysis (ASR), we test several advanced microphysics schemes and assess the impact of model resolution. Both the P3 and Morrison two-moment schemes (candidates for the next ASR version) overproduce the opaque mode, whereas the Goddard scheme used in ASR overproduces the transmissive mode. The opaque bias in P3 and Morrison arises mainly from excessive low-level, optically thick clouds over sea ice. Among all schemes, P3 best preserves the distinctiveness of the two radiative modes. Using this scheme, PWRF forced with end-of-century CESM1 output projects a shift toward more frequent opaque conditions, consistent with long-term observations at the North Slope of Alaska. While PWRF shows promise as a tool for dynamically downscaling climate model output, persistent cloud-related biases, especially over ice, warrant caution in future projections. Continued improvements in cloud representation are essential to obtain more quantitative insight into Arctic radiative regime changes.

#### 15 1 Introduction

Over the past several decades, the Arctic has exhibited amplified changes in sea ice cover (Stroeve and Notz, 2018), near-surface air temperature (Box et al., 2019), and precipitation (McCrystall et al., 2021), a phenomenon referred to as Arctic amplification (Previdi et al., 2021; Serreze and Barry, 2011; Rantanen et al., 2022). These enhanced changes arise from a combination of processes, including the ice–albedo feedback (Curry et al., 1995), the lapse-rate feedback (Boeke et al., 2021), and increased poleward energy transport (Cai, 2005). Another contributing factor is the rising frequency of optically thick clouds (Kay et al., 2016; Morrison et al., 2018; Tan and Storelymo, 2019).

Optically thick clouds act like a blanket, enhancing downwelling longwave radiation (DLR) at the surface. Periods of increased DLR contribute to surface warming and delay sea ice formation, which in turn fosters conditions favorable for further cloud formation due to increased moisture availability (Morrison et al., 2018; Arouf et al., 2024). The Arctic atmosphere fre-

<sup>&</sup>lt;sup>1</sup>Dept. Atmospheric and Oceanic Sciences, University of Wisconsin-Madison, Madison, Wisconsin, USA

<sup>&</sup>lt;sup>2</sup>Cooperative Institute for Meteorological Satellite Studies, Madison, Wisconsin, USA

quently alternates between prolonged (days to weeks) transmissive periods (clear or optically thin clouds with relatively low DLR) and opaque periods (optically thick clouds with relatively high DLR) (Morrison et al., 2012). This phenomenon leads to bimodality in distributions of DLR throughout the year (Bertossa and L'Ecuyer, 2024).

Long-term in situ observations from the North Slope of Alaska (NSA) indicate a substantial increase in DLR over the past 25 years, particularly in late autumn (Bertossa and L'Ecuyer, 2025; Riot-Bretêcher et al., 2025). In November, for example, DLR has increased by more than 30 W/m² since the turn of the century. This trend coincides with a near complete disappearance of clear-sky conditions and a corresponding shift from bimodal to more unimodal DLR distributions resembling the opaque radiative mode. These radiative changes are accompanied by significant declines in sea ice extent in the nearby waters of the Beaufort Sea and Bering Strait. While it is not possible to diagnose cause and effect from these observations, the implied feedbacks suggest that the evolving behavior of the transmissive and opaque states plays a critical role in shaping the broader Arctic climate system.

However, long-term observational records like those at NSA are scarce across the Arctic, making it challenging to assess whether similar shifts are occurring elsewhere. Additionally, the Arctic climate system is highly nonlinear, complicating efforts to project whether these trends will persist. To address these data gaps, researchers often turn to reanalyses, which integrate sparse observations with model output to 'complete' the climate state. However, conventional reanalyses such as MERRA2 and ERA5 struggle to represent the bimodal nature of DLR (Bertossa et al., 2025). They either produce an unrealistic intermediate mode or underrepresent the opaque regime due to deficiencies in cloud frequency and phase representation.

Similar cloud deficiencies are well-documented in global climate models (GCMs) (Kay et al., 2016; Taylor et al., 2019). Due to their computational demands, GCMs operate at relatively coarse spatial and temporal resolutions, relying heavily on parameterized cloud processes that are often not well-suited to represent polar clouds (McIlhattan et al., 2017; Cesana et al., 2012; Shaw et al., 2022). As a result, climate models are likely ill-suited for investigating the behavior of the two Arctic radiative states.

This limitation is illustrated in Fig. 1, which compares seasonal distributions of observed Arctic surface DLR with output from the popularly used Community Earth System Model version 2 (CESM2; Danabasoglu et al., 2020). In all seasons, the unique bimodal structure associated with the two preferred states is evident in observations. CESM2 exhibits large discrepancies, not only compared to the fully observed probability density function (PDF), but also in the mean of the distribution, consistently under-representing the more transmissive mode associated with clear-sky fluxes. The European Centre for Medium-range Weather Forecasts Reanalysis 5 (ERA5; Hersbach et al., 2020) product is also offered for comparison. Both the GCM and the reanalysis product struggle to capture the observed bimodality.

There are several commonly used methods to compensate for the resolution limitations of climate models, one being dynamical downscaling. Dynamical downscaling uses a limited-area, high-resolution model (sometimes referred to as a regional climate model, RCM) whose boundary conditions are driven from a global climate model (like CESM) to resolve smaller-scale processes (Giorgi and Mearns, 1991). RCMs can also be tailored to the unique conditions of specific regions. For example, the Polar-optimized Weather Research and Forecasting model (PWRF) is designed to better simulate polar processes, such as ambient aerosol concentrations and surface energy exchange over ice (Hines and Bromwich, 2008).

70

**Figure 1.** Comparisons of surface downwelling longwave radiation (DLR) for CESM2, ERA5 and ASRv2 to observations. Observations are determined from CloudSat-CALIPSO's 2B-FLXHR-LIDAR product which can sample surface fluxes with high accuracy (see text discussion). CloudSat was fully operational from 2007-2010, so only those four years for each product and dataset are shown for comparison. PDFs are built from gridboxes or observations which reside within this study's model domain (see Fig. 2a), native resolutions of each product are not altered (adapted from Bertossa et al., 2025).

Bertossa et al. (2025) finds that the Arctic System Reanalysis version 2 (ASRv2; Bromwich et al., 2018), an Arctic specific reanalysis built on PWRF, reproduces the two-state behavior of Arctic DLR better than other reanalyses, although some discrepancies remain. This is further illustrated in Fig. 1, where ASR exhibits a broader DLR distribution that more closely aligns with observations, particularly in spring and autumn, and even shows evidence of bimodality in autumn (Fig. 1d). These findings suggest that PWRF may have inherent advantages in capturing the Arctic's distinct radiative regimes, making it a promising candidate for studying their evolution under past, present, and future climate conditions.

The primary objectives of this study are to (a) evaluate the ability of PWRF as an RCM for investigating the behavior and evolution of surface radiative states in the Arctic, with particular focus on identifying and understanding model deficiencies and (b) use PWRF as an imperfect yet useful tool to predict the future evolution of Arctic radiative states. Analysis will focus on autumn where ASR performs notably better than CESM2 or ERA5 and Bertossa et al. (2025); Arouf et al. (2024) find enhanced changes in DLR correlated to the relative frequencies of opaque and transmissive states. While previous work has often focused on analyzing specific case studies or events (such as transitions between clear and cloudy states) few studies have examined whether models can accurately simulate the climatological PDF of DLR that emerges from the accumulation of such events, namely the observed bimodality.

Given the limitations of coarser resolution models, which tend to favor a nonphysical intermediate state, high-resolution tools like PWRF offer an opportunity to better resolve the Arctic radiation budget. We focus on the Atmospheric Radiation Measurement North Slope of Alaska (ARM NSA) site, where a significant increase in surface DLR in November over the past 25 years has coincided with a loss of the transmissive mode. By comparing PWRF output to observations at this site, we assess whether the model preserves the observed bimodal structure or exhibits the same biases seen in coarser-resolution models, offering insights into its reliability for simulating future Arctic radiative behavior.

Table 1. PWRF model configuration

| Parameter                 | Selection                                               |
|---------------------------|---------------------------------------------------------|
| Domain                    |                                                         |
| Horizontal Grid           | 12.5 km (d01) and 2.5 km (d02)                          |
| Dimensions                | 215 x 215 (d01) and 360 x 360 (d02)                     |
| Time step                 | 60 s                                                    |
| Number of Vertical Levels | 52                                                      |
| Model Top Pressure        | 25 hPa                                                  |
| Physics                   |                                                         |
| Microphysics              | Р3                                                      |
| Cumulus                   | Kain–Fritsch (d01)                                      |
| Radiation                 | Rapid Radiative Transfer Model (longwave and shortwave) |
| Planetary Boundary Layer  | MYNN 2.5                                                |
| Atmospheric Surface Layer | MYNN                                                    |
| Land Surface Model        | Noah Land Surface Model                                 |
| Boundary Conditions       |                                                         |
| Data                      | ERA5/ CESM1 Bias Corrected                              |
| Frequency                 | 6 hours                                                 |

## 80 2 Model Description and Sensitivity Analysis

90

Version 4.5.1 of Polar WRF (PWRF) is used for all model simulations in this study (Hines and Bromwich, 2008). Compared to standard WRF, PWRF incorporates enhanced parameterizations for ice, surface processes, and microphysics, making it better suited for polar environments. These modifications have demonstrated improved performance in simulating both Arctic and Antarctic conditions, including cloud properties (Hines et al., 2015; Xue et al., 2022). The model configuration used here employs a one-way nested grid with an outer domain at 12.5 km resolution and an inner domain at 2.5 km resolution. The inner domain explicitly resolves convective processes, meaning no cumulus parameterization is used. That being said, convection is generally less influential at high latitudes. While the chosen grid resolutions are somewhat arbitrary, they do not significantly impact the presented results (see discussion). Both domains are centered over Barrow, Alaska to facilitate comparisons with long-running observational data from the ground pyranometer.

Figure 2 presents observed distributions of DLR across the described PWRF domains, as well as at the ARM-NSA station, for the years 2007–2010. The domain-wide distributions are derived from CloudSat & CALIPSO's 2B-FLXHR-LIDAR product, which combines CloudSat's cloud profiling radar with CALIPSO's lidar to achieve high sensitivity to cloud conditions, including those near the surface. This synergy enables the retrieval of detailed cloud radiative properties and allows for a systematic assessment of the occurrence and spatial extent of the two dominant radiative states (Bertossa et al., 2025). Complimentary to the long-running single-point perspective offered by the ARM-NSA station, the satellite data provides broader

105

Figure 2. DLR and  $CRE_{LW}$  PDFs for 2007-2010 for the model domain used in this study. (a) The outer domain (d01, dashed), inner domain (d02, solid) and location of the ARM-NSA station (star), to which the two domains are centered on. (b-e) The seasonal distributions of surface downwelling longwave radiation using CloudSat-CALIPSO's 2B-FLXHR-LIDAR product for 2007-2010 for the outer domain (blue) and inner domain (orange). Distributions from ARM-NSA observations for the same years are also provided (black). (f-i) as (b-e) but for longwave cloud radiative effect.

spatial coverage that is critical for evaluating domain-wide radiative behavior. As shown in Fig. 2, the outer domain, inner domain, and NSA site all exhibit relatively similar DLR PDFs, with clear evidence of bimodality in each case and across seasons, highlighting the widespread and persistent nature of these radiative regimes.

Numerical models such as PWRF are highly sensitive to the choice of parameterization schemes, particularly for microphysics and boundary layer processes (Taylor et al., 2019; Inoue et al., 2021). Unfortunately, there is rarely a universally optimal configuration, different schemes may perform better depending on the dominant processes during a given simulation period. Running ensembles with multiple parameterization sets is often computationally infeasible, especially when simulating a pseudo-climatology over many days, as done here. Therefore, we leverage prior studies to identify an optimal set of parameterizations suitable for reproducing a wide range of Arctic conditions. Specifically, we adopt most parameter settings from the Arctic System Reanalysis (ASR) version 2 (Bromwich et al., 2018), an Arctic-specific reanalysis product built upon PWRF simulations. ASR has undergone extensive testing and mirrors many of the choices from other PWRF studies (Sledd and L'Ecuyer, 2019; Graham et al., 2019; Cho et al., 2020; Avila-Diaz et al., 2021). Table 1 outlines the parameter set used, for further details readers are directed to Bromwich et al. (2018) and the references therein.

To construct a November DLR climatology, we simulate the first and last five days of the month each year using hourly output. The first 12 hours of each run are discarded for model spin-up; increasing the spin-up period does not have a significant effect on the presented results. By focusing on just the first and final five days of November, we capture the range of DLR conditions (maximum and minimum) for November (Fig. A1) while also conserving computational resources. This approach enables simulations across more years, allowing a better representation of interannual variability.

125

130

While the majority of our parameterization options are consistent with those used in ASRv2, we investigate the use of a more advanced microphysics scheme to improve the representation of Arctic cloud processes. The Goddard scheme used in ASRv2 is acknowledged to be somewhat dated due to more recent advances in how super-cooled liquid is represented in clouds (Dodson et al., 2021). To assess potential improvements, we conduct sensitivity tests over a 5-day period in November that exhibits well-defined bimodality in both DLR and  $CRE_{LW}$  (Fig. 3). For this period, the Goddard scheme exhibits a clear underdevelopment of optically thick clouds and thus fails to capture the opaque DLR mode. This behavior is consistent with known ASRv2 biases, which systematically under-represents the opaque state (see Bertossa et al., 2025, and Fig. 1).

In contrast, both the Morrison 2-moment scheme and the Predicted Particle Properties (P3) two-moment scheme, each widely used in the modeling community (Morrison et al., 2005; Morrison and Milbrandt, 2015; Hines et al., 2019; Zou et al., 2021), produce a more realistic distribution, with two distinct DLR modes. Between these two options, the P3 scheme tends to exhibit more clearly separated modes, while Morrison shows slightly more intermediate behavior. Ultimately, we adopt P3 in part because it is also indicated as a candidate for the next version of ASR (see author discussion in Dodson et al., 2021), however we include some complimentary results from the Morrison scheme.

P3 uses a single ice category represented by four mixing ratio variables: total mass, rime mass, rime volume, and number. The main motivation behind the scheme is to omit the use of relatively arbitrary ice categories (i.e., 'ice' versus 'snow' versus 'graupel') in favor of a continuum. The P3 scheme is found to produce some of the best matches to observations, specifically for clouds containing super-cooled liquid water, which have a strong role in the evolution of the two Arctic states (Bromwich et al., 2009; Listowski and Lachlan-Cope, 2017; Hines et al., 2019; Cho et al., 2020).

**Figure 3.** Sensitivity analysis of PWRF microphysics and model configurations. Probability density functions are shown for a single 5-day simulation (November 1–6, 2008) using ERA5-forced boundary conditions. CloudSat-CALIPSO overpasses that intersect the d02 domain during this period are used as the observational reference (grey bar). Simulated PDFs are shown for the following configurations: the baseline model setup (Table 1, solid blue), increased vertical resolution with 75 model levels (dashed blue), the Morrison double-moment scheme in place of P3 (green), and the Goddard scheme in place of P3 (orange).

### 3 Representation of Current Climate

Before forcing PWRF with future conditions to explore the evolution of the states, it is important to first determine whether PWRF can accurately simulate observed conditions—specifically, the existence of two preferred states. To simplify this evaluation, we break up the problem into several elements that can be examined sequentially, initially setting aside the added challenge of assessing how well a climate model can produce realistic present-day forcing conditions necessary for the formation of these radiative states. We begin by driving PWRF with ERA5 reanalysis data (hereafter PWRF-ERA5). We present select case studies, evaluate model performance in a climatological context, test its sensitivity to resolution and domain size, and, finally, establish the impact of forcing the model with CESM present day climate simulations as opposed to ERA5.

## 140 3.1 Case Studies

Although ERA5 has its own limitations, it is generally expected to approximate observed conditions more closely than climate models because it assimilates observations to reconstruct the historical evolution of the atmosphere. In contrast, global climate models are often run unconstrained by observations, meaning their simulated atmospheric states may deviate significantly from the actual climatology. ERA5 is commonly used to provide boundary conditions for WRF simulations.

**Figure 4.** Example PWRF-ERA5 simulation for November matched to ARM-NSA observations. (a) Timeseries of surface downwelling longwave radiation for PWRF (blue) compared to ARM's pyranometer (orange). (b) Histograms of DLR built from (a). (c,d) as (a,b) but for longwave cloud radiative effect. The timestep depicted with the black vertical dashed line is examined in Fig. 5 and discussed in the text.

Figure 5. (a) Example 2d field generated with PWRF-ERA5 for November 1, 2008 at 14 UTC. Greys and blacks indicate the presence of clouds, while blues and reds indicate surface temperature. Thick yellow line indicates the ice boundary. The red line depicts an intersecting CloudSat & CALIPSO swath that occurred during this time. (b) The 2d cloud mask (black) and 1d  $CRE_{LW}$  (red dot) associated with the CloudSat swath according to the PWRF-ERA5 simulation. (c) as (b) but from true CloudSat-CALIPSO observations. (d) PDFs of DLR from the CloudSat-CALIPSO swath according to PWRF-ERA5 (blue) and CloudSat-CALIPSO (black bar). (e) as (d) but for  $CRE_{LW}$ .

Figures 4 and 5 present a sample PWRF simulation from November 1–6, 2008, comparing output from the 2.5 km inner domain (d02) to observations. Figure 4 shows a time series at the grid point nearest the ARM NSA site, allowing for direct comparison between the model and in situ observations. Overall, PWRF-ERA5 reproduces observed conditions reasonably well in this case. While some discrepancies are evident in CRE<sub>LW</sub> and DLR, the model accurately captures two regime transitions: a shift from an opaque to a transmissive state around 00 UTC on November 2, and a return to the opaque state around 0-12 UTC on November 3. Notably, both the model and observations show distinct bimodal behavior in DLR, with modes occurring at similar values and frequencies, which is ultimately the goal for PWRF in the context of this study.

To give spatial context of these regimes, Fig. 5 presents a 2D snapshot from 14 UTC on November 1 (indicated by the dashed line in Fig. 4), coinciding with a CloudSat and CALIPSO overpass. Observational data from the 2B-FLXHR-LIDAR product reveal low-level, liquid-containing clouds extending from approximately  $70.0^{\circ}$ N to  $73.2^{\circ}$ N along the swath. These clouds exhibit high  $CRE_{LW}$ , supporting the the larger DLR mode. PWRF also simulates low-level cloud cover in the same region, but the spatial extent is larger than what is observed.

Despite this overproduction, the model still captures the presence of two distinct DLR modes along this swath, with mode locations similar to those derived from the 2B-FLXHR-LIDAR observations (Fig. 5d). However, the increased cloud coverage leads to an overrepresentation of the opaque mode in the model's DLR distribution compared to observations. This behavior is somewhat unexpected, as all reanalysis products evaluated in Bertossa et al. (2025) tended to underproduce opaque conditions.

This result may suggest that while PWRF benefits from high-resolution and advanced microphysics, it could also be overly sensitive to cloud-forming processes under certain conditions.

**Figure 6.** 2007-2010 PDFs of DLR, DLR $_{clr}$  and CRE $_{LW}$  for d02 of the model study. PDFs are generated using CloudSat and CALIPSO 2B-FLXHR-LIDAR (gray bar), ERA5 hourly data (red dotted), ASR 3-hourly data (blue dotted), PWRF-ERA5 hourly data using the P3 scheme (red solid), and PWRF-ERA5 hourly data using the Morrison 2-moment scheme (green solid).

# 3.2 Climatology

To evaluate how well PWRF-ERA5 captures the two observed DLR modes in a more systematic and climatological context, Figure 6 presents November PDFs of DLR, clear-sky DLR (DLR<sub>clr</sub>), and  $CRE_{LW}$ , aggregated over four years (2007–2010), from the PWRF-ERA5 simulations. These are compared directly to CloudSat-CALIPSO observations, as well as to ASRv2 and ERA5 reanalysis products, over the same period.

A key feature in Fig. 6 is the systematic overproduction of opaque clouds in PWRF, especially when using P3– an issue already evident in the single swath example (Fig. 5). This overproduction results in a diminished lower (transmissive) DLR mode and an overrepresentation of the opaque mode in the PDFs. Specifically, clear-sky or optically thin clouds (CRE  $\approx$  0 W/m²) are observed roughly twice as often as they are simulated in PWRF (Fig. 6c), indicating a persistent bias in cloud fraction. Using the Morrison 2-moment scheme does better to produce additional transmissive scenes, though sparsely observed intermediate conditions are also more frequent, and thus, the two modes in DLR are less distinct than what is observed. These results align with a previous study in Nam et al. (2024), where P3 exhibits the highest cloud frequency bias (an overestimation relative to in-situ aircraft measurements) compared to four other microphysics schemes, including the Morrison 2-moment scheme.

Despite this overproduction of opaque scenes, PWRF more effectively preserves the bimodal structure of  $CRE_{LW}$  compared to 'raw' ERA5 reanalysis, which instead exhibits a blended intermediate mode. This intermediate behavior likely reflects weaker distinctions between transmissive and opaque regimes and could obscure important climate shifts, such as the observed shift towards more opaque conditions (Bertossa and L'Ecuyer, 2025). Crucially, capturing this bimodal structure in  $CRE_{LW}$  is a necessary requirement (though not the sole requirement) for reproducing distinct modes in DLR (Bertossa et al., 2025).

Most existing reanalysis products fail to meet this fundamental requirement, underscoring the need for continued development of models like PWRF if we are to reliably project future changes in Arctic radiative regimes.

It is interesting that PWRF with P3 demonstrates the inverse behavior of ASRv2. While ASRv2 retains a more distinct bimodal distribution compared to ERA5, it overproduces clear-sky conditions and underrepresents the opaque regime. Furthermore, using Morrison 2-moment better matches the observed cloud frequency, but lacks the distinct CRE modes evident when using P3. This emphasizes the sensitivity to the choice of microphysics schemes, not only to particular cases, but also when producing a climatology such as this. Since the opaque mode dominates in observations, one could argue that PWRF, with the P3 microphysics scheme, produces a more realistic simulation of Arctic longwave radiative states, despite still overproducing cloud cover.

One might speculate that the discrepancy in cloud cover arises because CloudSat and CALIPSO fail to detect the low-level clouds that PWRF accurately simulates. However, this is unlikely. The combined CloudSat-CALIPSO product is capable of detecting clouds at altitudes below 1 km and has been validated with numerous polar-specific studies (McIlhattan et al., 2017; Blanchard et al., 2021; Liu, 2022; Bertossa and L'Ecuyer, 2024). For example, as shown in Fig. 5, nearly all observed clouds during the overpass reside below 1 km, and many are confined to the lowest 500 m. Further confirmation comes from surface-based measurements at the ARM-NSA site, where DLR PDFs for the same time period are compared to the nearest PWRF grid point (Fig. A2). Here too, the model overproduces the higher DLR mode, consistent with an overabundance of cloud cover not solely attributable to detection limitations.

## 3.3 Influence of Model Resolution and Domain

Here, we examine the influence of grid resolution on the derived radiative flux distributions. Specifically, we perform an additional nested simulation from November 26 2007 to December 1 2007, introducing a high-resolution (500-meter) domain. Figure 7 shows the resulting PDFs of DLR and  $CRE_{LW}$  across the various grids.

In Figure 7(a, b), we compare the two primary domains used throughout this study (12.5 km and 2.5 km resolution), along with a PDF generated by bilinearly interpolating the coarser 12.5 km grid (d01) to the 2.5 km grid points (d02). The two domains exhibit notablely different PDFs over the five-day period. The interpolated coarse-grid data differs from the native 2.5 km output, especially for the  $CRE_{LW}$  distribution, suggesting that the higher-resolution domain is explicitly resolving additional physical processes, such as convection, which are not parameterized at this scale.

In contrast, the difference between the 2.5 km and 500 m grids is relatively minor. PDFs generated by interpolating the 2.5 km grid (d02) to the 500 m grid (d03) are nearly identical to those from the native d03 output, whereas interpolating from the coarser 12.5 km grid to d03 introduces more noticeable discrepancies. This suggests that refining resolution beyond 2.5 km yields diminishing returns for representing the processes that shape the longwave flux distributions.

This result is perhaps unsurprising, as the opaque cloud state is primarily linked to relatively homogeneous, large-scale stratus decks and thick ice clouds that are driven by synoptic-scale disturbances (Bertossa et al., 2025). These cloud structures tend to be metastable and are not strongly influenced by small-scale convective processes. However, as noted by Eirund et al. (2019), the Arctic is shifting toward more frequent occurrences of stratocumulus clouds, suggesting that cumulus convection may be-

Grid Resolution Comparison for 2007/11/26 - 2007/12/01 Sim.

**Figure 7.** Comparisons between DLR and  $CRE_{LW}$  PDFs from PWRF-ERA5 using different horizontal grid resolutions for November 26 - December 1, 2007. (a) DLR PDFs for d01 (blue solid), d02 (green solid), and d01 data interpolated to the d02 grid boxes (blue line). (b) as (a) but for  $CRE_{LW}$ . (c) DLR PDFs for d03 (red solid), d01 data interpolated to the d03 grid boxes (blue line), and d02 data interpolated to the d03 grid boxes (green line). (d) as (b) but for  $CRE_{LW}$ .

come increasingly relevant. This shift highlights the potential need for more explicit representation of convective processes in future high-resolution Arctic modeling efforts.

## 3.4 Cloud Fraction Bias

Why does PWRF overproduce clouds in this configuration? A key factor appears to be how the model handles cloud formation over different surface types. In particular, discrepancies in cloud occurrence between cold, ice-covered surfaces and relatively warmer open ocean likely contribute to the departure from observations. Figure 8 compares vertical profiles of observed cloud fraction and  $CRE_{LW}$  distributions over ocean and ice surfaces to those simulated by PWRF-ERA5.

Consistent with prior observational studies (e.g., Morrison et al., 2018; Arouf et al., 2024), satellite observations indicate that cloud occurrence is generally lower and less opaque over ice-covered surfaces than over open ocean. This leads to a higher frequency of scenes with  $CRE_{LW}$  near 0 W/m<sup>2</sup>, corresponding to clear-sky or optically thin cloud conditions. In contrast, PWRF simulates much higher near-surface cloud fractions over both surface types, reaching nearly 95% over sea ice (approximately 20% higher than observed) and 93% over open ocean (about 5% higher). These values represent substantial overestimations relative to observed frequencies, with cloud representation over sea ice being particularly unrealistic.

One potential explanation for this overproduction is improper coupling between the lowest atmospheric model level and the surface, especially over ice. For example, the cold surface may induce condensation at the lowest model level, forming persistent, optically thick low clouds that are not supported by observations and exert an unrealistic radiative effect. This behavior is especially pronounced in simulations using the P3 microphysics scheme, where cloud fractions near the surface (below 1 km) are much greater than observed. Such issues reflect broader challenges in polar modeling, where accurately representing boundary layer processes and microphysics remains difficult (Tjernström et al., 2008; Vavrus and Waliser, 2008; McCusker et al., 2023).

Despite these limitations, PWRF does capture some important characteristics of the observed radiative regimes. Notably, it reproduces the relative difference in  $CRE_{LW}$  between clouds over ocean and those over sea ice. As expected, clouds over ocean surfaces tend to be optically thicker (due to enhanced surface moisture fluxes) which increases emissivity and  $CRE_{LW}$ . While PWRF overpredicts the overall cloud frequency, the position of the  $CRE_{LW}$  modes for each surface type aligns well with observations when using the P3 scheme, suggesting that the model captures some key physical relationships correctly.

Differences between the P3 and Morrison microphysics schemes become especially evident in this context. Over ocean, both schemes behave similarly in terms of cloud frequency (93% for P3 and 95% for Morrison) and  $CRE_{LW}$  distribution. However, over sea ice, notable distinctions arise. Both schemes simulate high cloud frequencies (93-96%), but only the P3 scheme maintains the distinct opaque  $CRE_{LW}$  mode seen in CloudSat and CALIPSO observations. In contrast, the Morrison scheme produces more optically thin  $(CRE_{LW}$  near 0 W/m²) and intermediate  $(CRE_{LW}$  between 10 and 80 W/m²) clouds, leading to a weaker representation of the opaque mode. This distinction helps explain why Morrison has a less exaggerated opaque mode compared to P3, despite similarly high cloud fractions (see Figs. 3 and 6).

#### 3.5 Forcing PWRF with a Climate Model

A bias-corrected form of CESM1 has been created for the specific purpose of forcing WRF (Monaghan et al., 2014). The applied bias correction is derived from ERA-interim reanalysis and is meant to correct the mean state towards the observed seasonal climatology, while still maintaining the variability produced by CESM (Bruyère et al., 2014). This dataset is widely used by the dynamical downscaling community (e.g., Krayenhoff et al., 2018; Ghanbari et al., 2023).

Figure 9 shows November distributions of surface DLR, clear-sky DLR (DLR $_{clr}$ ), and longwave cloud radiative effect (CRE $_{LW}$ ) for near-present (2006-2010, black lines) simulations using CESM1 as the boundary forcing. These runs, hereafter referred to as PWRF-CESM, provide a means to evaluate how PWRF performs when driven by a climate model rather than a reanalysis product such as ERA5. Although DLR bimodality remains absent, the CRE $_{LW}$  distributions closely resemble

**Figure 8.** (a) Vertical cloud fraction over sea ice for 2007-2010 according to PWRF-ERA5 simulations using the P3 scheme (red), the Morrison 2-moment scheme (green), and CloudSat-CALIPSO observations (grey line) for d02. (b) Longwave cloud radiative effect distributions over sea ice for 2007-2010 for d02; the legend indicates cloud frequency. (c,d) as (a,b) but over ocean.

those from PWRF-ERA5, with two distinct modes still evident (albeit at frequencies still differing from observations). The differences in DLR distributions between the two simulation sets likely arise because variability in PWRF-CESM reflects that of CESM rather than being constrained to match observations like ERA5 (e.g., note the larger spread in the DLR<sub>clr</sub> distribution compared to PWRF-ERA5). These results suggest that PWRF, when coupled with CESM, can provide valuable insight into how Arctic radiative states may evolve under different forcing conditions. However, quantitative interpretation of these changes will remain limited until further improvements are made to PWRF that correct the cloud overproduction.

#### 4 Predicted Evolution of Arctic Radiative States

The CESM1 dataset includes a 20th century simulation with the observed forced response from 1861-2005, followed by three different Representative Concentration Pathway (RCP) scenarios (4.5, 6.0, and 8.5) until 2100. Although CESM produces ensembles to capture the internal variability of the Earth system, only one ensemble member has the complete three-dimensional fields required to force WRF (member #6, known as the 'Mother of All Runs'; MOAR). We use this member to explore how surface DLR PDFs evolve over time in PWRF.

**Figure 9.** November PDFs of DLR, DLR<sub>clr</sub>, and CRE<sub>LW</sub> for d02 of the model study using PWRF-CESM and the P3 scheme. Three different 5-year periods are simulated: 1980-1984 (blue), 2006-2010 (black), and 2095-2099 (red).

In addition to the near-present baseline, Figure 9 depicts November distributions for a historical period (1980–1984) and a late-century projection (2095–2099) under the RCP6.0 scenario. The DLR distribution shows a clear positive shift by the end of the century, driven primarily by increases in clear-sky forcing associated with warmer, moister conditions in the future Arctic atmosphere. As previously mentioned, this signal is consistent with observed and expected behavior in a rapidly warming and moistening Arctic, lending confidence that PWRF-CESM captures some key aspects of the changing Arctic climate.

There are modest changes in the  $CRE_{LW}$  distributions across the three periods. In the historical period, the opaque cloud mode is shifted toward slightly lower values compared to the present and future periods, consistent with clouds having cooler emission temperatures and/or reduced emissivities. By the end of the century, this mode shifts to slightly larger values and becomes more frequent, suggesting a further increasing frequency of optically thick clouds. Bertossa and L'Ecuyer (2025) observe that the transmissive mode was largely present prior to the turn of the century in the western Arctic, in contrast the the historical period produced by PWRF-CESM. This suggests that the opaque cloud frequency bias identified in ERA5-forced simulations also persist when CESM output is used as the boundary forcing. Consequently, while PWRF-CESM projects an increase in DLR, the magnitude of this response is likely underestimated, as the loss of the transmissive mode should amplify the increase.

Nevertheless, using CESM1 as boundary conditions for PWRF provides a promising foundation for future work. Importantly, the framework still captures the presence of the two observed  $CRE_{LW}$  modes, offering a physical basis for exploring their evolution. With targeted improvements, particularly reducing low-level cloud biases and refining the representation of cloud

radiative properties across surface types, this setup could be used to explicitly examine how the two preferred states are changing and to assess their broader impact on the Arctic climate.

#### 5 Conclusions

Accurately capturing the preferred radiative states of the Arctic is a challenge for models (Bertossa et al., 2025), and PWRF is no exception. In this study, we evaluated how well several configurations of PWRF reproduce the observed bimodal distribution of downward longwave radiation (DLR) in November, a month that has been particularly sensitive to changing Arctic conditions (Bertossa and L'Ecuyer, 2025). We tested a range of microphysical schemes and boundary forcing datasets. While further refinement is needed, our results show that PWRF demonstrates promise, it captures the location of cloud radiative effect ( $CRE_{LW}$ ) modes in good agreement with observations, though it consistently misrepresents their frequency. Specifically, opaque clouds are present too frequently compared to what is observed. Given the dominant role of these radiative states in influencing the Arctic surface energy balance, better representing their frequency and structure is essential for improving our understanding of Arctic climate in general. Targeting the factors that lead to the states' misrepresentation offers a pathway to enhance the overall performance of PWRF in Arctic applications.

By adopting a setup similar to that of the operational version 2 of the Arctic System Reanalysis, we assess the potential impact of using a more advanced microphysics scheme. Specifically, we compare the performance of the P3 and Morrison 2-moment schemes, candidates for version 3 of ASR, to the Goddard scheme used in ASRv2. Our findings reproduce the known tendency of the Goddard scheme to overproduce the transmissive state while underrepresenting opaque cloud occurrences. Conversely, replacing it with the P3 scheme results in the opposite issue, with an overrepresentation of the opaque state. The Morrison 2-moment scheme does a better job at exhibiting bimodal DLR distributions but lacks distinctive modes in  $CRE_{LW}$ , frequently producing unphysical intermediate  $CRE_{LW}$  conditions. One could say that Morrison is getting closer to the observed bimodal DLR distributions compared to P3, but for the wrong reasons.

Related to this, we identified a systematic overproduction of clouds, particularly over sea ice, that is inconsistent with observations. This suggests that PWRF is overly sensitive to surface conditions, producing clouds too readily in cold, stable regimes where observed cloud occurrence is suppressed. Improving this behavior may require revisiting the boundary layer parameterization or microphysics scheme rather than focusing solely on increasing vertical or horizontal resolution. Explicitly, we found that while increasing horizontal resolution from 12.5 km to 2.5 km has some impact on the DLR distribution, further refinement to 500 m does not significantly alter the results. Furthermore, increasing the vertical resolution did little to improve the performance in the tested simulations (Fig. 3). This indicates diminishing returns in resolution improvements for this problem.

Importantly, we demonstrated that PWRF, when forced by ERA5, preserves the distinction between the two  $CRE_{LW}$  modes better than 'raw' ERA5 radiative flux output. This reinforces PWRF's value as a downscaling tool for improving Arctic cloud and radiation representation. Moreover, PWRF's flexibility in accepting a range of boundary forcing datasets opens opportunities for climate change assessments. For example, we applied this PWRF set up with CESM1 MOAR output to simulate

historical, near-present, and future DLR distributions. While the PWRF-CESM simulations inherit the same biases, such as an underrepresentation of clear-sky conditions, PWRF-CESM also exhibits the two dominate  $CRE_{LW}$  modes, which enables investigation of how these states may evolve under projected climate forcing. If model physics can be further refined, especially to better capture clear-sky conditions, then PWRF could serve as a powerful tool for studying how Arctic cloud regimes and radiative states will respond to ongoing climate change.

## **Appendix A: Sampling Strategy**

Are observations of DLR from the first and final five days of November representative of November as a whole? Here we use a bootstrapping approach with both the ARM-NSA and the CloudSat-CALIPSO daytime record to answer this question. For each observation dataset, 5-year consecutive November DLR distributions are generated using the entire month of observations (black line) and only the first and final five days (red line). In both cases, the limited sampling strategy closely aligns with the full November sampling PDFs, giving indications that the limited sampling used for the model study is sufficient to capture November variability when compiled over enough years.

# Influence of Model Sampling Strategy

**Figure A1.** Evaluation of the model sampling strategy. (a) The mean DLR PDF across the different 5-year consecutive Novembers using full sampling from ARM-NSA (black line) and using only observations from the first and final five days, mimicking the model study (red line). Grey shading depicts one standard deviation surrounding the mean. (b) as (a) but using CloudSat-CALIPSO data spanning d02.

**Figure A2.** 2007-2010 PDFs of DLR for ARM-NSA (grey) compared to the nearest PWRF-ERA5 grid point. Simulations using both the P3 (red) and Morrison 2-moment scheme (green) are shown.

Author contributions. CB performed the analysis and drafted the manuscript. TL and DH provided guidance on the analysis and contributed to manuscript revisions.

Competing interests. The authors state that there are no competing interests.

Data availability. Data were obtained from the Atmospheric Radiation Measurement (ARM) user facility, a U.S. Department of Energy (DOE) Office of Science user facility managed by the Biological and Environmental Research Program. The North Slope of Alaska surface radiation measurements used in this study is publicly available through the ARM Data Center via Data Discovery (DOI: 10.5439/1227214) and can be accessed with a free ARM user account. ERA5 data used here is publicly available via the Copernicus Climate Change Service (C3S) Climate Data Store. Results for particular simulations can be delivered upon request.

Acknowledgements. This work was supported, in part, by the NASA Earth Ventures-Instrument (EV-I) program's Polar Radiant Energy in the Far-Infrared Experiment (PREFIRE) mission under grant #80NSSC18K1485. Some of the work was conducted at the University of Wisconsin-Madison for the Jet Propulsion Laboratory, California Institute of Technology via NASA grant #G-39690-1. The authors would also like to thank the CloudSat Data Processing Center for providing the satellite data products that complemented the basis of this study.

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
