# Peer review of "Past, Present, and Future Arctic Radiative States Simulated by Polar-WRF"

_EGUsphere, 2025_

## Author Comment (AC1)

**General comments:**

The study is well presented and nicely constructed. The findings are interesting. I suggest some minor/specific comments.

**Specific comments:**

- L-63 : not in DJF, why?

  - ASR tends to struggle with producing a sufficient number of opaque clouds in general (see Bertossa et al., 2025). DJF is the season when the opaque mode is least prevalent in observations, as ice-covered surfaces and relatively low atmospheric moisture inhibit the formation of clouds with high water path. As a result, any underrepresentation of the opaque mode in ASR leads to an effective loss of the bimodal structure during DJF.

- Figure 2: How is the CRE obtained from the ground station? Is it the same definition as the satellite? (F_dwn – F_up)_all - (F_dwn – F_up)_clr ? The maximum of occurrence ARM-NSA Vs CSC does not occur at the same value of CRE-Lw (75W/m2 Vs ~65W/m2). It looks like the satellite retrieval occurs at higher values for all seasons and agrees the most in SON. Can the author explain why?

  - While it's the same basic formula, there are important differences with how each component is derived. The ground-based CRE relies on direct measurements of downwelling radiative fluxes (F_dwn) and the clear-sky (and CRE) fluxes are calculated from measured water path values. That is, F_dwn is known and F_dwn_clr is inferred. Whereas the satellite-based F_dwn (and CRE) is a derived quantity computed using a radiative transfer model that combines clear-sky flux estimates with retrieved hydrometeor properties. The close agreement between these two independent approaches provides strong confidence in the existence of the identified cloud states in the first place. Differences in the magnitude of the dominant CRE mode (i.e., 65 versus 75 W/m2) likely arise from a combination of factors, including the differing calculation methodologies, but also colocation differences. In particular, CSC values are based on satellite overpasses within a 1x1 degree box surrounding the station. Because the station is located near the coast, whether an overpass samples sea ice or open ocean can substantially influence the retrieved CRE, contributing to the observed offsets.

- L-175. Why is P3 overproducing opaque clouds while the Morrison one the intermediate mode? I mean, what is the microphysical configuration that causes

these discrepancies? Figure 6 shows that almost all the DLR clr have the same distribution across the different datasets/configurations and the differences are in the CRE-LW.

○ This mainly comes from differences in cloud water path distributions. Below we've provided an example from a randomly selected simulation set. We've added vertical lines at 0.1 g/m2 (values lower than this are almost always transmissive according to observations, see Bertossa et al. 2025) and 30 g/m2 (values higher than this are almost always opaque according to observations). Values in between these lines can lead to "intermediate" CREs, though these don't actually occur frequently in nature. We see that Morrison has a greater relative frequency in this intermediate zone compared to P3. If we were to plot LWP and IWP specifically (below), we see it can be attributed to both phases.  Beyond this, what physical mechanism causes P3 to produce more opaque clouds than Morrison hasn't been determined here. Our main goal for this study was to determine the scheme that best matches observations, rather than investigate the root mechanisms that lead to this. That being said, see our response to reviewer 2 #4 for a possible explanation to why P3 produces more opaque clouds.

[Figure]

- L-190: interesting because climate models usually underestimate opaque clouds. Wonder what in P3 scheme creates the overproduction of opaque clouds.

  - See previous response for a possibility.

- L-193. CALIPSO can get fully attenuated in opaque clouds, and CloudSat suffers from the strong echo from the surface that can alter low cloud detection over rough surfaces, but shouldn't be an issue over the Arctic. Is the issue the coarse vertical resolution of the satellites?

  - While this is a good consideration, the coarse resolution has little effect on the actual CRE in this case, so long as the cloud is detected in the first place. Cloud detection rates by CloudSat-CALIPSO product are very high in these cases (Bertossa and L'Ecuyer 2024). Henderson et al. (2013) find that the radiative influence of extending the cloud base has a relatively small effect (1.5 W/m2 / 250m). So even if the base can't be accurately captured due to attenuation or surface clutter, the effect on CRE should be relatively small. We have added note of this in the revised manuscript. This has been added to the revised manuscript (Lines 192-197).

- Figure 7: Did the author do some sensitivity studies on the vertical resolution of the radiative computation? Maybe it can explain some of the discrepancies. Or authors can add uncertainty bars for the different datasets and check whether they overlap.

  - Sensitivity to vertical resolution for the radiative computation is partly looped into Fig. 3 (p3 versus p3_hv). Where, increasing the number of vertical levels by ~50% does very little to change the resulting CRE distribution (both in terms of what clouds are produced, and how the radiative effect of those clouds are calculated). We have added explicit mentions of the lowest model level for each vertical configuration to the revised manuscript (Fig 3 caption and model configuration table).

- Figure 8: The mean cloud fraction profile in the Morrison scheme (over ice) shows a bimodal distribution with a large population of low clouds (~500m) and mid-level clouds (5-6 km). I don't understand why we don't observe a bimodality in the CRE-LW distribution that follows the cloud fraction one. Could the authors clarify this point?

  - This is great to point out, and part of the reason why doing these model evaluation studies can be difficult. "Clouds" in models and clouds detected

by observations are not necessarily equivalent. "Clouds" in models are often defined by a non-zero water path value, which doesn't necessarily have a significant radiative effect. Furthermore, observations do indicate that there are preferred water path values (see Bertossa et al. 2025 Fig. 9&10). That is, there is a physical instability that pushes the state towards near-zero water path values or very high water path values that cause the CRE to saturate. There are a lot of great suggestions as to why this may be the case (see for example Morrison et al 2012 and references therein). This plot indicates that the Morrison scheme does not appropriately lead to these preferred water path groupings as we see in observations (and P3). See discussion in Lines 240-250.

**Figure comments:**

- Using green and red in the same plot is not color blind friendly. I suggested changing the line style when these two colors are used.

  - Revised as suggested. Green bars have been replaced with black.

- In fig. 6, the legend says 'Obs' for the black line. I suggest kipping CSC for satellite retrieval and ARM-NSA for the ground station (or similar) throughout the paper. Same for the simulations, keeping the same notation can be helpful.

  - Revised as suggested. Obs has been replaced with CSC here.

---

## Author Comment (AC2)

Some PWRF setting questions:

1. 52-vertical level is used in this paper, is this number enough, what is the lowest model level? Like ASR v2 is 71 levels, with lowest level at 4m. I don't think re-run any model is necessary, but I would suggest authors provide more info in the table 1. PWRF default (at PMG) is 71 levels, with more levels at lower altitude. That is designed to better capture near surface conditions, and cloud formation.

   We tested higher vertical resolution in Figure 3 by increasing the number of model levels to 75. In the 52-level configuration, the lowest model level is located 12 m above ground level, whereas in the 75-level configuration the lowest level is at 4 m AGL. Increasing the vertical resolution has a negligible impact on the resulting CRE distributions. We have now explicitly listed the lowest model level in the model configuration table and clarified this point in the discussion accompanying Figure 3.

2. I might miss it. Is unified Noah or Noah MP is being used here?

   We are using Unified Noah here. We have marked this more explicitly in the model configuration table. We found that, in our use cases, Noah MP tended to lead to instabilities in the Polar WRF model.

3. Is there any nudging applied? If so, what variables for what levels?

   The full set of boundary conditions are nudged towards the ERA5 reanalysis data every 6 hours (see model configuration table). However, the domains are freely evolving. We mention this explicitly in the revised manuscript (Lines 82-86).

4. This is not a suggestion, but more like a discussion. P3 (option 50) in general provides pretty good results, especially related to super-cooled liquid water in clouds. However, we also notice sometimes it can produce large value of cloud LWP (e.g., like unrealistic large for a short period of time at WAIS compared to obs. While Thompson & Morrison did a better job). I believe Hines et al. 2019 also kinda mentioned that. I am wondering if authors encounter similar issue in Arctic.

   Yes, we observed a similar issue in some cases. That said, because LW cloud radiative effects saturate relatively quickly with increasing water path (beyond ~30 g/m2, further increases in water path have little impact on CRE), this was less problematic for our analysis, which focused on comparing CRE distributions. However, this behavior may be a key reason for the overproduction of opaque cloud states, as clouds that should maintain lower LWP can unrealistically grow too rapidly.

5. I am not familiar with ERA5 default surface type for Arctic regions (aka whether it is good enough). For Antarctic region, ERA5 use a quite old land surface cover (which still have Larsen A & B ice shelf included...). Thus, a more up-to-date land-ocean description is usually needed (Like REMA). For high-resolution simulations, more accurate SST and SI observations are usually recommended to be included as initial fields. As authors mentioned in Ln 220, difference surface type matters. I am wondering if more information can be provided here regarding the simulations has been done in this study.

We did not test a more advanced land-ocean surface representation. Our rationale was that ERA5 is widely adopted and provides a convenient "plug-and-play" option for Polar WRF (whose development is more niche than standard WRF). That said, Figure 8 somewhat ironically suggests that, in its current configuration, the model is largely insensitive to surface conditions, despite observations indicating that the frequency of these cloud states is strongly controlled by them. As a result, incorporating a dataset with a more accurate surface representation would likely require additional development of the atmosphere–surface coupling to get any benefit. This development effort is beyond the scope of this study.

6. There is no need to add any simulations. I am wondering authors have ever tested Thompson (aerosol aware) by any chance?

Yes, we tested it briefly. We found that its behavior was similar to the Goddard scheme, in that the cloud states were less distinctly separated and tended to be somewhat blended. For reference, we include below an example from a simulation we still have archived for 1 October 2007, comparing results from P3, Thompson, and Goddard.

[Figure]

Minor:

Ln 15-20: Several research suggested a poleward shift of atmospheric river activities for Arctic region, which will enhance the transport of moisture, energy and warmth. I think this is worth mentioning as background change for Arctic region.

*Zhe Li, Qinghua Ding ,A global poleward shift of atmospheric rivers.Sci. Adv.10,eadq0604(2024).DOI:10.1126/sciadv.adq0604*

We have included this citation in our discussion of increased poleward energy transport in the Arctic

Introduction structure:

The Intro is in a good shape in general. This is just a suggestion. Introduction section has a decent number of short paragraphs, I am wondering authors ever think about merging some of those, to deliver the key messages more clearly.

As suggested, we have merged some of the shorter paragraphs into one.